# De-fine: Decomposing and Refining Visual Programs with Auto-Feedback

## ABSTRACT

Visual programming, a modular and generalizable paradigm, integrates different modules and Python operators to solve various vision-language tasks. Unlike end-to-end models that need task-specific data, it advances in performing visual processing and reasoning in an unsupervised manner. Current visual programming methods generate programs in a single pass for each task where the ability to evaluate and optimize based on feedback, unfortunately, is lacking, which consequentially limits their effectiveness for complex, multi-step problems. Drawing inspiration from benders decomposition, we introduce **De-fine**, a training-free framework that automatically decomposes complex tasks into simpler subtasks and refines programs through auto-feedback. This model-agnostic approach can improve logical reasoning performance by integrating the strengths of multiple models. Our experiments across various visual tasks show that **De-fine** creates more accurate and robust programs, setting new benchmarks in the field. The anonymous project is available at https://anonymous.4open.science/r/De-fine_Program-FE15

## CCS CONCEPTS

• **Computing methodologies** → **Computer vision problems**.

## KEYWORDS

Visual Programming, Training-free Visual Processing, Task Decomposition, Program Refinement

## 1 INTRODUCTION

Large visual models [2, 7, 21, 26, 36] based on transformers, excel in various visual tasks, such as object detection [20], visual question answering [4], video description [37], and visual reasoning [35], by using large-scale unsupervised pre-training and supervised multi-task training. However, these end-to-end models do not essentially reveal internal logic and need fine-tuning for new tasks [27], which might be costly and challenging for complex and long-tailed tasks. In pursuit of task inference without additional training, interpretable programming-based approaches have been developed, allowing the expression of logic and reasoning for visual tasks through the assembled code modules. Previous works like Visual Programming [8] and ViperGPT [32], use code-generation models to compose vision-language models (VLMs) [39, 41] into subroutines and assemble a program for visual tasks. ViperGPT, for instance, uses a provided API to access modules and generate executable code. It requires no further training and leverages the expressive power of programming languages, making it effective for solving complex visual tasks.

Despite their advantages, current programming-based methods tend to generate lines of atomic code sequentially in a single pass, without properly decomposing the task into smaller manageable

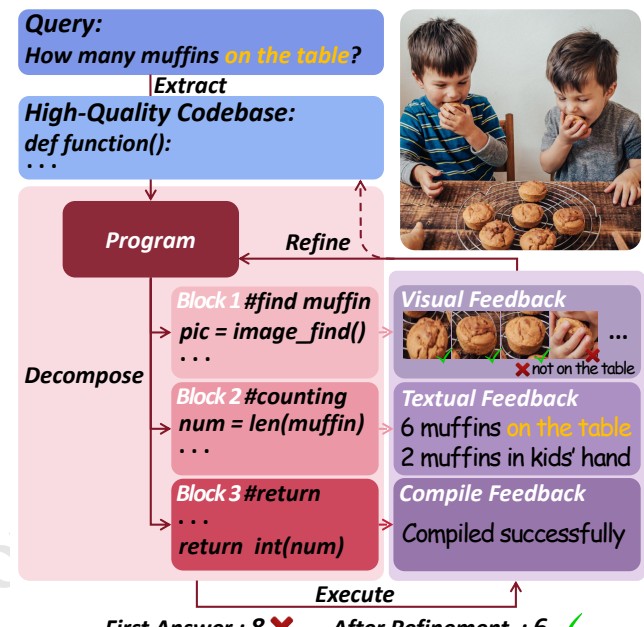

**Figure 1: De-fine decomposes the tasks into executable program blocks and automatically refines the program based on multifaceted feedback from the execution.**

subtasks and generating corresponding program blocks separately. Such a manner leads to two main issues: 1) **Insufficient hierarchical task decomposition**: Failing to plan the program structure in a hierarchical manner, previous works [8, 32] largely struggle to handle complex tasks efficiently, especially for handling logically intricate tasks. This could potentially lead to sub-optimal or hard-to-maintain code and contradict the original design intention for the compositional task. Further, without a clear decomposition of the problem, identifying and fixing bugs becomes daunting as the error could be deeply embedded within intertwined program logic. Modular code, in contrast, enables easier isolation and resolution of issues. 2) **Ineffective intermediate feedback utilization**: Since the program is generated in a single pass, programming-based methods fail to take advantage of intermediate results and system-returned values during code execution that can enhance code quality and facilitate debugging. This indicates a lack of adeptness in leveraging real-time feedback and adjusting code logic dynamically, which could otherwise lead to more refined and debuggable code outputs.

Drawing an analogy from the process of human programmers [9, 24], we consider four essential steps typically followed in program development: 1) **Reference**: search for the most relevant algorithm that aligns with the given task's logical structure to serve as a high-level logical reference. 2) **Decomposition**: based on the reference's structure, decompose and frame the programming task into several

subtask modules, later constructing an executable draft program. 3) **Feedback**: obtain systematical feedback for program revision by comparing the expectation with program output, intermediate variables, and compiler return values. 4) **Iteration**: progressively refine the program based on the feedback until the desired correct outcome. We exemplify the idea in Figure 1. For the query "How many muffins on the table?", one may first retrieve a similar query (*e.g.* "How many toys on the desk?") and parse its logic structure as a reference. Then we decompose the task into two subtasks: "find the muffins on the table" and "count the number of muffins". The programmer will further evaluate the output, intermediate results, and the program against the expectation, and finally refine the program iteratively until satisfied.

Inspired by this process in software engineering, we propose `De-fine`, a training-free framework that **decomposes intricate tasks** into executable program blocks by modeling the logic structure of relevant tasks and **automatically refines the program** based on multifaceted feedbacks from the execution. Our framework advances in relieving humans from the tedious process of converting ideas into programs. Specifically, as shown in Figure 2, `De-fine` generates an abstract logical prompt that reveals the internal task logic without redundancy. This prompt is selected based on semantic similarity to the task query and can imitate the logic of the program after masking. Then, prompted by abstract logical examples, a large language model (LLM) generates executable programs that decompose the tasks for the queries. After execution, `De-fine` automatically refines the program blocks by multifaceted feedback derived from the program results, intermediate variables, and compiler messages. These systematic feedbacks are summarized by categories through multiple targeted specific models.

Importantly, the two core modules of `De-fine` collaborate well with each other. Mutually, **the refinement part** can extract applicable codes based on feedback and expand the codebase to a logically well-structured one which will be used as prompts for future tasks. This logical codebase even does not require manual annotation or any ground-truth programs for training. Reversely, **the decomposition part** also contributes to the refinement part by instructing the program to generate more detailed code that provides richer reference information for feedback. These two components are interdependent and synergistically reinforce each other.

Our empirical evaluation on various benchmarks reveals that the proposed method is able to effectively decompose intricate multi-step problems and be adapted to various visual tasks such as grounding [14], reasoning [35], and image question answering [1, 12, 23]. By capitalizing on multi-faceted feedback and the capabilities of other multi-modal language models, we achieve state-of-the-art (SOTA) zero-shot results on five tasks without model fine-tuning or supervised training.

Overall, our contributions are as follows:

- We revisit visual programming as a task of modular programming and optimization through feedback, solving them by the software engineering principles. With `De-fine`, we break down tasks into executable program blocks and refine them automatically using multifaceted feedback.
- `De-fine` constructs an abstract logical prompt to sufficiently preserve the internal logical reasoning structure of the draft

program, and systematically defines four types of feedback to optimize the quality and performance of programs.
- Without any supervised training data, `De-fine` achieves SOTA zero-shot performance on tasks such as image question answering, visual reasoning, and grounding.

## 2 RELATED WORK

Program generation and self-optimization have seen renewed momentum owing to the incredible understanding and generation capabilities of LLMs. We now discuss previous program generation, recent work in using LLMs for vision, and advances in self-refinement.

**Program Generation**. Visual program generation is an active research domain that aims to synthesize programs performing vision tasks using neural symbols [28] or Python modules [19, 33]. This approach is based on the assumption [3, 13] that vision tasks are inherently compositional and can be decomposed into atomic perceptual units like lines of code. Yet, complex tasks pose a challenge for this approach, as the generated code is often sub-optimal due to the insufficient semantic understanding of LLMs [15]. In contrast, `De-fine` can generate well-performance programs with a hierarchical structure.

**Visual Programming with LLM**. Programming-based methods [29, 38, 40] are scalable and interpretable for vision tasks, as they can incorporate any vision or language module with a predefined interface. Additionally, they enable fine-grained image processing and editing through code-level operations. The progress of the program generation model [19] enhances the synthesis of programs for visual tasks without task-specific training. Nevertheless, they still require multiple manually labeled codes as context-learning examples. While `De-fine` can automatically retrieve relevant examples to assist in program generation.

**Refinement with Auto-feedback**. Even for human programmers, there is no guarantee that the code written on the first try is always accurate. Therefore, we hope the model can provide multifaceted feedback and refine its previously generated program. Previous work like Self-debug [5] uses the error message of the program as feedback to modify the code generated. Self-refine [22] optimizes the output through feedback and refinement iteration. However, this feedback comes from a single modality of text and is only generated from the final result. In contrast, `De-fine` can provide feedback on variables during code execution and generate feedback types for different variable types, enabling it to handle visual, textual, and error messages.

## 3 METHODOLOGY

To address the limitations of current programming-based approach in insufficient decomposing and ineffective utilizing feedback, we propose `De-fine`, a training-free framework for decomposition and refinement, which eliminates the necessity to fine-tune any existing pre-trained models. Our method is shown in Figure 2, given a task query and a visual input, `De-fine` first generates an abstract logical prompt that guides the decomposition and well-structured program synthesis. (Section 3.1). During execution, it generates multifaceted feedback considering the program results, intermediate variables, and compiler messages (Section 3.2). `De-fine` then automatically

**Step1: Abstract Logical Prompt Construction**

*Qurey:*

*How many wheels does the cars have?* —Retrieve→

*Abstract Logical Prompt:*

```
#step1:Find the car
#step2:...
def function(image):
    #step1:Find the object <pad>
    image_patch = ImagePatch(image)
    <pad> = image_patch.find("<pad>")
    #step2:return the number of <pad>
    return len(<pad>)
def function(image):
    ...
```

*Codebase*

←Extract

*Visual input*

**Step2: Program Generation and Execution**

*Generated Program:*

```
def function(image):
    #step1:Find the car
    image_patch = ImagePatch(image)
    car_patches = image_patch.find("car")
    #step2:Find the wheel
    num_cars = 0
    num_wheels = 0
    for car_patch in car_patches:
        wheel_patches = car_patch.find("wheel")
        num_wheels += len(wheel_patches)
    #step3:return the number of wheels
    return num_wheels
```

**Step4: Code Evolution**

```
def function(image):
    #step1:Find the car
    image_patch = ImagePatch(image)
    car_patches = image_patch.find("car")
    #step2:Count the number of wheels
    num_wheels = 0
    for car_patch in car_patches:
        is_isetta = car_patch.simple_qurey
                    ("Is this car a BMW isetta ?")
        if bool_to_yesno(is_isetta):
            num_wheels += 3
        else:                        After Refinement : 6 ✓
            num_wheels += 4
    #step3:return the number of wheels
    return num_wheels
```

←Update

*De-fine*

**Step3: Multifaceted Feedback Generation**

*Visual Feedback:*

*VLM: This is a BMW isetta that features only three wheels*

*VLM: This is a blue BMW isetta with only three wheels*

...

*Textual Feedback:*

num_wheels = 0

...

num_wheels = 3

*LLM: The program calculated two cars and three wheels*

*Human Feedback (optional):*

*Human: Some wheels cannot be seen from a specific perspective*

*Compile Feedback:*

*Code Static Analysis: Unused variable 'num_cars'*

*IDE: Successful Compilation!*

**Figure 2: De-fine is a programming-based framework that can decompose tasks and refine the program. We summarize the process into four steps: (1) De-fine first constructs an abstract logical prompt. (2) We generate the program and execute it. (3) During execution, De-fine automatically generates multifaceted feedback for optimizing. (4) De-fine keeps the well-optimized code based on feedback and expands the codebase for future use. The pseudocode algorithm is shown in Appendix A.**

refines the program according to the systematic feedback to produce a well-performing code (Section 3.3). Moreover, these self-improved high-quality programs enrich the codebase for future use (Section 3.4). The whole process does not require any additional input and relies solely on the self-optimization of the model.

## 3.1 Abstract Logical Prompt Generation

In the context of visual programming, an effective decomposition necessitates fulfilling two criteria: logical problem decomposition and hierarchical code organization. Absent these, the code generated without explicit intention guidance may result in ambiguity or even compilation errors. To address this challenge, we initially dissect the query into natural language sub-steps, providing an explicit **logical step**. Subsequently, we extract relevant code from a database to refer to its logical structure, forming an implicit **abstract code**. These steps culminate in the creation of the **abstract logical prompt**. The prompt significantly improves code synthesis by eliminating irrelevant variables and enhancing the visibility of the internal logic, offering a substantial advantage in the clarity and effectiveness of program generation.

**Logical Step Generation.** For a given query $q$, we leverage the zero-shot reasoning capabilities of the LLM to decompose it into sub-steps $Q = \{q_1, q_2, \ldots, q_N\}$ where $q_i$ represents the $i$-th step in addressing the original problem. This logical step will be

embedded into the generated code as comments in the future to provide readability and explicit intentional guidance for the code generation engine.

**Abstract Code Extraction.** Given a textual query $q$, we first retrieve $K$ code snippets $Z = \{z_1, z_2, \ldots, z_K\}$ where $z_i$ represents the $i$-th code whose description is similar to $q$ from a codebase $B$ (for details, please refer to Section 3.4). After getting candidates, we use placeholders (<pad>) to discard the irrelevant parts (*e.g.* variable names, specific conditions in *if* statements) and obtain the abstract code $\hat{Z} = S(Z)$, where $\hat{Z} = \{\hat{z_1}, \hat{z_2}, \ldots, \hat{z_K}\}$. This abstract code does not require meticulous design. By masking irrelevance that is not pertinent to the current task, it guides the model to focus on learning universal solutions or strategies and directs the code generation engine to decompose the tasks hierarchically.

For integration, we find the intuition from previous studies [10, 11] that codes with similar natural language descriptions share an analogous logical structure. We go one step further by sorting abstract code based on text similarity to the substeps in the logical step. We construct an **abstract logical prompt** $AL = \{Q, \hat{Z}\}$ shown in Figure 3, enabling program generation $z = \pi(q, AL)$ via a generator $\pi$ (*GPT-3.5-Turbo*). During generation, the model is instructed to insert comments derived from the logical step, thereby enhancing the extraction and feedback of pertinent information.

**Figure 3: The pipeline of abstract logical prompt generation. Initially, we generate sub-steps to address the given query. Subsequently, we retrieve the most relevant code based on the semantic relevance of code comments and substeps. Then, we mask any irrelevant or redundant information in the retrieved code. This AL is finally provided as a prompt to the code generation model.**

## 3.2 Multifaceted Feedback Generation

During program execution, `De-fine` takes advantage of intermediate results and system return values during code execution to enhance code quality and facilitate debugging. To achieve this, we systematically define several types of feedback: 1) Visual Feedback, 2) Textual Feedback, 3) Compile Feedback, and 4) Human Feedback (*optional*) that can dynamically adjust code logic based on the execution outcomes. These multifaceted feedbacks are generated by corresponding feedback generators, leading to more refined and debuggable code outputs.

Specifically, after getting the program $z$, we apply an execution engine $\phi(z, x)$ and a feedback generator $G(\phi)$ to execute $z$ on the input image $x$. The generator $G$ extracts intermediate variables $V = \{v_1, v_2, \ldots, v_n\}$ (*e.g. image patch, string, comment*) and generates feedbacks $F = \{F_{visual}, F_{textual}, F_{compile}, F_{human}\}$ :

**Visual Feedback**: The execution of the grounding and finding functions in the program $z$ return image patches with corresponding bounding boxes. We use a VLM (*mPLUG-Owl* [39]) to process the intermediate image variable, generating feedback for dual objectives: 1) *Image caption extraction*: The VLM captions image $x$ and the image patches in $V$ with bounding boxes, converting visuals to text to clarify ambiguities in code generation queries. 2) *Sub-step verification*: It verifies whether the image patches in $V$ match the expected results of each substep in $z$. For example, if a substep is supposed to crop a face, the VLM checks whether the cropped image contains a face or not and generates Visual Feedback accordingly.

**Textual Feedback**: We use the cognitive capability of the language model (*LLaMA* [34]) to provide Textual Feedback for two purposes: 1) *Logical question answering*: We ask the model to answer logical questions about the text output, like how the intermediate

variables $V$ deduce the final answer, and whether they match the substep reasoning process. 2) *Text summarization*: For atomic code, the program may produce verbose and repetitive output strings if there is a loop in the program. For the entire program, we also need to verify whether the reasoning between the steps in the comments is correct. We summarize these string outputs by the language model and generate Textual Feedback from a higher level.

**Compile Feedback**: We first conduct a static analysis to identify potential hazards, including variable shadowing and naming conflicts. During compilation, the compiler identifies and reports any syntax or semantic errors, such as omitted semicolons, undeclared variables, and data type incompatibilities. These notifications serve as Compile Feedback for refining the compilation process and improving the precision of code execution.

**Human Feedback (optional)**: `De-fine` may also cooperate feedback directly from humans for optional. In programming, users can iteratively modify the program alignment with their intentions until get the anticipated outcome. This intention termed Human Feedback, tends to be markedly explicit and facilitates human-in-the-loop inference. `De-fine` is capable of leveraging human knowledge and expertise, alongside human creativity and heuristic reasoning, to provide high-quality feedback. We show an example of human feedback in Figure 4d.

The feedback generated by `De-fine` above can consolidate the correct steps, clarify the ambiguous parts, and correct the wrong parts simultaneously. This enables the exploitation of the program itself and its intermediate variables, providing multifaceted feedback for the automatic refinement process of the model. We list the prompts required by the model in Appendix D.

**Table 1: Visual grounding task results. We report the accuracy of the REC (referring expression comprehension) task and *testA split* on the RefCOCO and RefCOCO+.**

|  | IoU(%) | |
| --- | --- | --- |
|  | RefCOCO | RefCOCO+ |
| GLIP [18] | 55.0 | 52.2 |
| ReCLIP [30] | 58.6 | 60.5 |
| GENOME [6] | 69.2 | - |
| ViperGPT [32] | 73.1 | 67.9 |
| **De-fine** | **75.2** | **70.0** |

## 3.3 Automatic Code Refinement

In the previous step, we obtained feedback that can integrate information from various sources, such as patches and strings from intermediate variables, result output from the program, and returns from the compiler, into new programs. This feedback is then used to optimize the draft program with the help of **De-fine**'s refiner, which is especially useful for code optimization, as it can enhance the performance and logic of the program.

The automatic refiner reuses our previous code generation model $\pi$. Given a query $q$, an initial program $z$, and feedback $F$, the refiner $\pi$ refine a new program $z^* = \pi(q, z, F)$ that improves on $z$ in terms of accuracy, conciseness, and logic. The execution engine $\phi$ then takes an input image $x$ and the refined program $z^*$, then generates a result $r = \phi(x, z^*)$ as the final output.

Unlike rule-based or heuristic-based strategies, our refinement is feedback-oriented, facilitating adaptability across diverse queries and programs through the assimilation of execution feedback. Additionally, it adopts a holistic perspective, thereby optimizing the program in its entirety rather than optimizing isolatedly. Moreover, the process is interactive and iterative, rather than being constrained to a one-off or static execution, thereby excelling in the enhancement of the program via multiple iterations of feedback from the user or the execution.

## 3.4 Codebase Evolution

**De-fine** takes advantage of optimized code that has a consistent and hierarchical structure refined by feedback. This type of code can produce optimal results and facilitate code reuse. By adding the optimized code back to the codebase, we can enhance the quality and reliability of the programs in the codebase over iteration.

We revisit the mention presented in Section 3.1 here. During the inference process, if there is no code corresponding to the current query in the codebase, the generated program will be added to it. Otherwise, it means that the updated code needs to be compared with the draft one. At this time, we execute them with auto-feedback as we mentioned in 3.2 and reuse the refiner as a selector and retain one of them.

**Initialization of the database**, we randomly select 20% of all samples within the current *testing* task to fabricate query-code dyads for code retrieval and generate the program first (**De-fine** will still infer them later). It is worth noting that **De-fine** is a training-free framework. Consequently, we do not utilize any training data from any tasks, aiming to circumvent potential misunderstandings or accusations of unfairness that might arise from

**Table 2: GQA Results. We report accuracy on the GQA *test-dev* set. $GT - data$=ground-truth data, $AL$=abstract logical prompt.**

|  | Accuracy(%) | $GT - data$ | Voting |
| --- | --- | --- | --- |
| VISPROG | 50.5 | ✓ | ✓ |
| BLIP-2 [16] | 44.7 | ✗ | ✗ |
| GENOME [6] | 45.6 | ✗ | ✗ |
| ViperGPT [32] | 49.7 | ✗ | ✗ |
| ViperGPT+$AL$ | 52.2 | ✗ | ✗ |
| **De-fine** | **55.3** | ✗ | ✗ |

**Table 3: Visual question answering and reasoning tasks results. We measure the accuracy (%) of the models on the *val set* of OK-VQA, the *test set* of TallyQA, and the *test set* of NLVRv2.**

|  | Accuracy(%) | | |
| --- | --- | --- | --- |
|  | OK-VQA | TallyQA | NLVRv2 |
| VISPROG | 52.6 | 68.1 | 62.4 |
| BLIP-2 [16] | 45.9 | 48.4 | - |
| Flamingo [2] | 50.6 | - | - |
| ViperGPT [32] | 52.5 | 70.2 | 62.9 |
| ViperGPT+$AL$ | 54.8 | 71.7 | 64.0 |
| **De-fine** | **57.1** | **73.2** | **67.3** |

using training set data or labels. Our approach is meticulously designed to avoid any form of data leakage. Hence, we only used the questions from the **test dataset (without labels)** as part of our initial dataset to further this goal. We first infer a subset of the testing set, thereby excluding the utilization of training set data. This procedure will eschew annotations for the filtration, as well as abstain from leveraging data from its training dataset, ensuring no data leakage risk. Subsequently, the feedback generated during execution will be utilized to filter these pairs and preserve executable codes. In this way, the codebase can be expanded as samples are continuously inferred, providing more accurate retrieval and more relevant results in subsequent iterations.

## 4 EXPERIMENTS

The **De-fine** framework exhibits the capability to execute a multitude of visual tasks without training or access to ground-truth data. In this section, we present the experimental setup (Section 4.1) and undertake a comprehensive evaluation across three distinct visual tasks: (1) visual grounding (Section 4.2), (2) compositional visual question answering (Section 4.3), and (3) zero-shot reasoning on image pairs (Section 4.4). Additionally, we have conducted extensive ablation studies to assess the impact of various parameters and components within our framework (Section 4.5).

## 4.1 Experimental Setup

**Model Setup**. For abstract code extraction, we can replace variables, judgments, and constant strings with placeholders (<pad>) in static code analysis. In practice, during the experiment, we use an end-to-end sketcher [17] to implement the mask operation. For the program generator, refiner, and filter, we adopt the *GPT-3.5-Turbo*

**Table 4: Ablation results (%) of individual components.**

|   |   | GQA | OK-VQA |
|---|---|-----|--------|
| 0 | Backbone | 49.1 | 52.0 |
| 1 | + decompose (in-context prompt) | 50.6 | 52.9 |
| 2 | + abstract logical prompt | 52.5 | 54.1 |
| 3 | + feedback | 54.8 | 56.2 |
| 4 | + code evolution | 55.2 | 56.7 |
| 5 | ViperGPT + in-context prompt | 51.0 | 53.5 |

*(1106)* [25] language model, as provided through the official OpenAI API. For visual feedback, we use *mPLUG-Owl (7B pre-trained version)* [39] as a caption extraction and logical judgment tool. For textual feedback, we utilize *LLaMA (7B)* [34] to extract the text output and intermediate variable returns in the program and generate corresponding feedback. In quantitative experiments, Human Feedback was not incorporated into our experiment due to efficiency and cost.

**Baselines**. We use the following baselines: Visual Programming [8], the procedure is given 24 contextual instances, run 5 times per execution for majority voting. For a fair comparison, all experiments utilize *GPT-3.5-Turbo (1106)* as the code generation engine within ViperGPT [32]. Furthermore, the analysis encompasses BLIP-2 [16], Flamingo [2], GLIP [18], ReCLIP [30] , GENOME [6] as additional comparative frameworks.

## 4.2 Visual Grounding

We compare different models on visual grounding tasks on Ref-COCO and RefCOCO+ [14] datasets. This task requires the model to locate the object in an image that corresponds to a given natural language description, as well as to demonstrate the ability of the model to reason about spatial relations and visual features.

As the result shown in Table 1, **De-fine** achieves a significant improvement over existing models under zero-shot settings. A potential reason for the inferior performance of end-to-end models (*e.g.* GLIP, ReCLIP) is their lack of an explicit representation of the internal reasoning structure and their inability to leverage modular tools. In contrast, ViperGPT can access modules via a predefined API, but **De-fine** surpasses it by automatically optimizing the program and refinement. This suggests that **De-fine** can validate and modify the generated program for better performance based on the feedback from intermediate variables at the refining stage. Furthermore, this result verifies the flexibility of **De-fine**, which can adapt to different queries and tasks by adjusting the program structure and parameters.

## 4.3 Compositional Visual Question Answering

**De-fine** is a novel method specifically designed for complex visual question answering tasks. It can intuitively show the process in which a complex problem is decomposed step by step and continuously improved program by its own feedback. In this section, we demonstrate the effectiveness of our model on three datasets: GQA [12], OK-VQA [23], and TallyQA [1].

As shown in Tables 2 and 3, programming-based methods, despite employing large foundation models, are constrained by their single-pass approach. Different from Visual Programming which

**Table 5: Ablation results (%) of multifaceted feedback.**

|   |   | Feedback Module | | | Accuracy | |
|---|---|--------|---------|---------|-----|--------|
|   |   | Visual | Textual | Compile | GQA | OK-VQA |
| 0 | Backbone |  |  |  | 52.6 | 54.4 |
| 1 | + Visual | ✓ |  |  | 54.5 | 56.3 |
| 2 | + Textual |  | ✓ |  | 53.9 | 55.0 |
| 3 | + Compile |  |  | ✓ | 52.9 | 54.8 |
| 4 | + Visual + Textual | ✓ | ✓ |  | 55.1 | 56.7 |
| 5 | + Visual + Compile | ✓ |  | ✓ | 54.8 | 56.6 |
| 6 | + Textual + Compile |  | ✓ | ✓ | 54.2 | 55.5 |
| 7 | **De-fine** | ✓ | ✓ | ✓ | 55.3 | 57.1 |

takes a majority voting strategy for maximum consensus predictions per query, **De-fine** decomposes the task into finer-grained subtasks without using any ground-truth data and incorporating feedback from multiple variables in iteration. Despite using the same abstract logical prompt as an example for ViperGPT, **De-fine** consistently outperforms existing models by a large margin in all VQA tasks.

**De-fine** can also perform error correction on the generated programs. By using the Compile Feedback that we designed, the model can rapidly identify and fix the flaws in the program in the subsequent generation.

## 4.4 Zero-shot Reasoning on Image Pairs

We extend **De-fine** to accomplish reasoning on multiple images, not just one. Our model performs well on the NLVRv2 [31] benchmark which involves verifying statements about image pairs, the results are shown in Table 3.

Current visual models can process multiple images as input, yet they treat each image in isolation. The interrelation of different images relies on network fusion, which is affected by the sequence and quantity of images. **De-fine** synthesizes information from diverse modalities via feedback and offers a comprehensive correction proposal. This enables the model to improve its performance on multi-image tasks substantially.

## 4.5 In-Depth Analysis

**Qualitative Analysis**. Figure 4 showcases how **De-fine** dynamically refines programs by systematical feedback across various modalities. We also display some error cases and generated feedback in Appendix E and F. A notable advantage of our approach is its facilitation of **human-in-the-loop programming**, enabling direct incorporation of human reasoning and knowledge through feedback. This process fosters a collaborative environment where users impart heuristic insights to the model, which in turn, validates these inputs via programmatic reasoning and outputs, enhancing human-computer interaction.

**Effectiveness of Individual Components**. We perform an ablation study in four configurations of our model on the GQA and OK-VQA tasks (Table 4): 0) backbone: only single-pass program generation and execution, 1) backbone + in-context prompt: with decomposition module and the in-context prompt that uses the retrieved sample directly 2) backbone + abstract logical prompt:

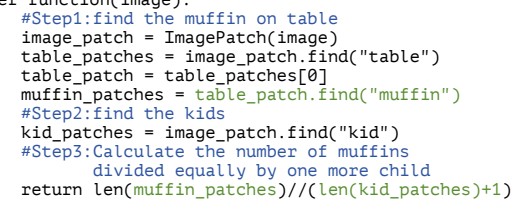

**Query: How much is it, if $1 for a bottle of Coke?**

**ViperGPT**
```python
def function(image):
    image_patch = ImagePatch(image)
    coke_patches = image_patch.find("coke")
    unit_price = 1
    cost = len(coke_patches) * unit_price
    return cost
```

**Visual Feedback:**
**Buy one get the 2nd one half price**
...

**Textual feedback:**
**The total cost is $3**
...

**De-fine**
```python
def function(image):
    #step1:Find all coke patches
    image_patch = ImagePatch(image)
    coke_patches = image_patch.find("coke")
    #step2:Define unit price and half price
    unit_price = 1
    half_price = 0.5
    #step3:Count the number of coke
    unit_price_coke = ceil(len(coke_patches) / 2.0)
    half_price_coke = floor(len(coke_patches) / 2.0)
    #step4:Calculate total cost and return
    cost = unit_price_coke * unit_price +
            half_price_coke * half_price
    return cost
```

**(a) De-fine can solve complex question answering tasks**

**Query: If there comes another kid,**
**how many muffins on the table can each kid have equally ?**

**ViperGPT**
```python
def function(image):
    image_patch = ImagePatch(image)
    muffin_patches = image_patch.find("muffin")
    kid_patches = image_patch.find("kid")
    return (len(muffin_patches)+1)//len(kid_patches)
```

**Visual Feedback:**
**A brown muffin on the table ×6***
**A muffin in a kid's hand ×2***
...

**Textual feedback:**
**We find 2 children, but the query stated that there is one more.**
...

**De-fine**
```python
def function(image):
    #Step1:find the muffin on table
    image_patch = ImagePatch(image)
    table_patches = image_patch.find("table")
    table_patch = table_patches[0]
    muffin_patches = table_patch.find("muffin")
    #Step2:find the kids
    kid_patches = image_patch.find("kid")
    #Step3:Calculate the number of muffins
            divided equally by one more child
    return len(muffin_patches)//(len(kid_patches)+1)
```

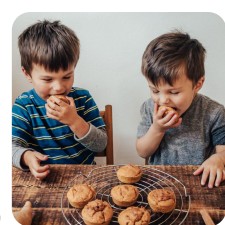

*\* Since each image receives a caption in Visual Feedback, for simplicity, we show it in the figure with a ×6 and ×2 notation.*

**(b) De-fine can modify program logic errors**

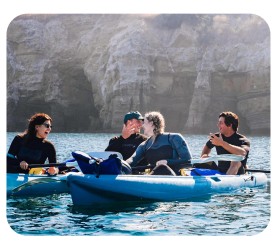

**Query: Find all the people on the front island.**
**(*no island in image at all )**

**ViperGPT**
```python
def function(image):
    image_patch = ImagePatch(image)
    island_patches = image_patch.find("island")
    island_patches.sort
    (key=lambda island: island.compute_depth())
    people_patches = island_patches[0].find("man")
    return people_patches
```

**Compile Feedback: IndexError: list index out of range**

**De-fine**
```python
def function(image):
    #Step1:find the island
    image_patch = ImagePatch(image)
    island_patches = image_patch.find("island")
    #Step2:sort and select the island in front
    island_patches.sort
    (key=lambda island: island.compute_depth())
    #Step3:if If there is no island, return "no island"
    if len(island_patches) = 0:
        return "There is no island in picture"
    #Step4:return all the people on the front island.
    people_patches = island_patches[0].find("man")
    return people_patches
```

**(c) De-fine can correct code compilation errors**

**Query:**
**According to the current clock, what time is it in Paris now?**
**ViperGPT**
```python
def function(image):
    image_patch = ImagePatch(image)
    clock_patches = image_patch.find("clock")
    time = clock_patches[0].simple_query
    ("what time is it in Paris now?")
    return time
```

**Human Feedback:**
**The picture was taken in London, while the query is about Paris**

**De-fine**
```python
def function(image):
    #step1:find the current clock and get time
    image_patch = ImagePatch(image)
    clock_patches = image_patch.find("clock")
    time = clock_patches[0].simple_query("what time is it?")
    #step2:Query the time difference between the two places
    time_difference = llm_query
    ("What is the time difference between London and Paris")
    #step3:Calculate the final time
    time_in_Pairs = llm_query("{time} {time_difference}")
    return time_in_Pairs
```

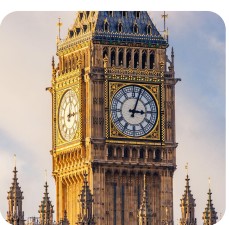

**(d) De-fine can update code with knowledge from human feedback**

**Figure 4: Refinement examples by the feedback of De-fine.**

with decomposition module and the abstract logical prompt, 3) backbone + decompose + feedback: with feedback generation and refinement of program, and 4) backbone + decompose + feedback + code evolution: with codebase updating component for better searching results. 5) We also compare our approach with ViperGPT given by the in-context prompt. The result provides strong evidence that the performance improvement is not attributable to the use of in-context prompts, but rather to the logical structure format of the abstract logical prompt.

By analyzing the data in the table, we conclude that 1) the decompose module enables effective decomposition of the task into multiple sub-tasks. 2) A non-redundant prompt truly guides the model to focus on logical imitation, which results in a significant improvement in solving complex problems. 3) Refinement by feedback provides the most enhancement, as feedback conveys high-level information and allows the model to revise and update the code which can correct the potential errors or integrate information to obtain the correct answer. 4) By code evolution, the model can

**Table 6: Impact of code generation engines.**

| | Accuracy(%) | | |
|---|---|---|---|
| Engine | GQA | OK-VQA | Average |
| Code-Llama (34B) | 54.7 | 56.6 | 55.65 |
| Code-Llama (70B) | 54.8 | **57.3** | 56.05 |
| GPT-3.5-Turbo (1106) | **55.3** | 57.1 | **56.20** |

**Table 7: Evaluation of Code Quality. We use pylint as a tool to evaluate code quality and report the average score and compilation success rate.**

| | Pylint score | Compilation accuracy |
|---|---|---|
| ViperGPT | 6.64 | 86.5% |
| De-fine (Code-Llama) | 7.21 | 90.2% |
| De-fine (ChatGPT) | **7.50** | **92.8%** |

accumulate more experience stored in the codebase and utilize it for future problem-solving.

**The Impact of Multifaceted Feedback**. Table 5 delineates the ablation studies conducted on varied feedback modules. Within this framework, Visual Feedback exhibited the most significant impact. This is attributed to the capability of visual feedback to amalgamate visual cues with textual information during code synthesis, offering a substantial enhancement for language models lacking direct access to visual data. Succeeded by Textual Feedback, providing ancillary support in the integration of context and semantics into the generated code. Ultimately, Compile Feedback, while beneficial for refining code style, appears to have a negligible effect on the overall accuracy. This phenomenon is ascribed to the high rate of successful code execution in the baseline, thereby limiting the scope for significant accuracy improvements through compilation feedback alone. To prove the improvements are not solely due to providing image captions or text information we conducted an additional ablation experiment in Appendix B.

**The Impact of Code Generation Engines**. We conducted experiments on different code generation engines, shown in Table 6. It is observable that variations in model and parameter size exert limited influence on `De-fine`, substantiating that our model-agnostic approach can be widely applied across diverse models. The programs generated by the code generation engine are presented in Appendix G, with an analysis of structure and style.

**Evaluation of Code Quality** The performance on VQA tasks directly indicates the code's ability to be correctly interpreted and executed. However, acknowledging the missing systematic evaluation for code generated within the VQA domain, we employ Pylint, a static code analysis tool. This tool is instrumental in evaluating the code quality produced by both ViperGPT and `De-fine`. The findings from this comprehensive analysis are detailed in Table 7. Remarkably, the code generated by our system achieved an average score that surpasses that of ViperGPT. We attribute this superior performance to the minimal presence of dependency issues and the elimination of redundant variables in our code. These improvements are in line with the observations from our other ablation studies, underscoring our system's refined code generation capabilities.

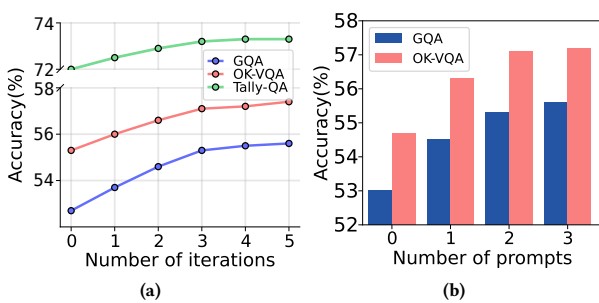

(a)  (b)

**Figure 5: Analysis on (a) the number of iterative refinement and (b) abstract logical prompts.**

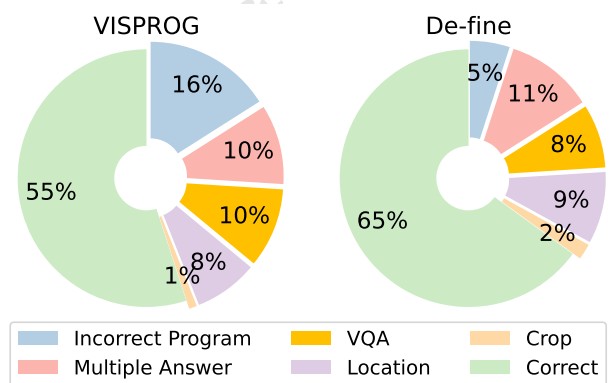

**Figure 6: Sources of error in GQA task.**

**Analysis on the Number of Iterative Refinement**. We show how the number of iterative refinements affects the performance in Figure 5a. The plot indicates that three iterations are sufficient to optimize the program, as the performance plateaus after that. Considering the token cost of accessing a large model, we adopt the outcomes of three iterations as our results.

**Analysis on the Number of Prompts**. To investigate how the number of abstract examples affects the performance, we varied the number of abstract codes (0-3) as prompts for the code generation model. Figure 5b shows that 2 program examples are sufficient to showcase the capability of the model, so we adopt the two program examples setting for our demonstrations.

**Human Evaluation**. To conduct an error analysis, we followed Visual Programming and manually selected 100 samples from GQA to identify the sources of errors, shown in Figure 6. The results indicate that our method outperforms Visual Programming in reducing the "Incorrect Program" errors. We will detail the types of errors and comparisons in Appendix C.

## 5 CONCLUSION

We propose `De-fine`, a method that decomposes tasks by constructing an abstract logical prompt to guide the well-performing code generation. After execution, `De-fine` refines the program based on the four types of systematic feedback. Through experimentation, we demonstrate that `De-fine` serves as a self-optimization approach that is model-agnostic, scalable, and interpretable.

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
