# OpenReview forum: "De-fine: Decomposing and Refining Visual Programs with Auto-Feedback"
_acmmm.org/ACMMM/2024/Conference — MM2024 Oral_

### Official Review · Reviewer_htG8 · 2024-05-24

**Rating:** 3
**Confidence:** 4

**Summary:**

This paper focuses on the visual programming framework and proposes a method to improve the quality of code generation in visual programming by building a high-quality code base and feedback on executing procession, thereby improving the effectiveness of the visual programming framework on various visual tasks.

**Strengths:**

1. This paper proposes a training-free improvement method for visual programming, a new and powerful multi-visual task framework, improving its performance by improving code quality.
2. The ablation experiment in this paper is very comprehensive, including a variety of visual programming frameworks and a variety of code generation engines.
3. Code error is a very common error in visual programming frameworks. It is good to improve the performance of visual programming by solving this problem.
4. The paper is well-written and reader-friendly.

**Limitations:**

1. Feedback may not be correct. Will it lead to an accumulation of errors? As noted in Figure 6, modules of vision programming can also have errors. Likewise, feedback models such as LLaMa also have errors.
2. Can the diversity of reference codes be guaranteed? Visual programming itself is a framework for multiple different tasks. One of its great attractions is the creativity of the code, and then the use of VLM combination to achieve multiple different tasks. But is it reasonable for De-fine to retrieve reference code logic in a limited code base? I hope the author can give some analysis of the diversity of the code base.
3. When De-fine executes the program, it needs to go through multiple rounds of feedback to give the final result. Will this lead to a serious increase in time overhead? (The visual programming framework itself has a slower execution speed, including code generation and calls to different visual modules)
4. My biggest problem is the motivation of this work. Although improving the quality of generated code by referring to existing high-quality code is a method that is very consistent with software engineering (line 130), code error is also a very common type of error in visual programming frameworks (statistical results in Figure 6 ). But for such a method, will it affect the generalization of the visual programming framework? When this method encounters new tasks, will the reference in the code base have a negative impact? For example, the current code base is based on the VQA task and the Grounding task, will it have an impact on its performance when reasoning on image pairs is encountered? Therefore, although improving the quality of code generation is a good direction to improve the visual programming framework, whether this approach will have an impact on the generalization of the original method remains to be evaluated.

Others, but not limitations:

In Table 7, I suggest that the code generation engines used by Vipergpt are also indicated (in the Vipergpt author's official code, ChatGPT should be used, so that there can be a more intuitive comparison with line 3 in Table 7).

I may need more explanations for my questions. I may consider changing my score.

**Suitability:**

3

---

### Official Review · Reviewer_gVLF · 2024-05-24

**Rating:** 5
**Confidence:** 2

**Summary:**

This paper introduces De-fine, a framework for visual programming that decomposes and refines visual programs using auto-feedback. The method leverages a large language model to generate programs based on an abstract logical prompt, and then refines the programs using feedback from program results, intermediate variables, and compiler messages. The proposed approach is model-agnostic and scalable, and can be applied to various visual tasks such as grounding, reasoning, and image question answering. Experimental results show that De-fine achieves state-of-the-art zero-shot performance on several benchmarks without any supervised training data.

**Strengths:**

1、Innovative and practical methodology: The paper presents a novel approach to visual programming that combines program decomposition and refinement using a large language model. The method is simple yet effective, and can be applied to a variety of visual tasks.

2、Insightful empirical findings: The experimental results demonstrate the effectiveness of the proposed method, showing that it achieves state-of-the-art zero-shot performance on several benchmarks. The paper also includes ablation studies that provide insights into the contribution of each component of the framework.

3、Well-structured review of relevant literature: The paper includes a comprehensive review of related work in visual programming, end-to-end models, and interpretable models. This review helps to contextualize the proposed method and highlights its unique contributions.

**Limitations:**

1、Inadequate implementation details for reproducing the study: While the paper provides a high-level overview of the proposed framework, it lacks detailed information about specific implementation choices. For example, it would be helpful to know more about how the abstract logical prompt is constructed and how the feedback is summarized by categories through multiple targeted specific models.

2、Limited evaluation and ablation studies for the proposed method: Although the paper includes some ablation studies, they are limited in scope and do not fully explore the effect of each component on the overall performance. More thorough evaluation and ablation studies would help to better understand the strengths and weaknesses of the proposed method.

3、Lack of clarity in exposition: Some parts of the paper could be made more clear and concise. For instance, the description of how De-fine generates an executable program based on an abstract logical prompt could be improved by providing a concrete example. Additionally, the explanation of how De-fine refines the program blocks using multifaceted feedback could be made more precise.

**Suitability:**

2

---

### Official Review · Reviewer_4WEC · 2024-05-29

**Rating:** 6
**Confidence:** 4

**Summary:**

This paper introduces De-fine, a training-free framework that decomposes complex visual tasks into executable program blocks and refines them using multifaceted feedback. Inspired by software engineering principles, De-fine creates an abstract logical prompt to guide the generation of programs and employs four types of feedback to optimize program quality. The framework achieves SOTA zero-shot performance on various visual tasks without requiring fine-tuning or supervised training.

**Strengths:**

- A training-free framework capable of performing visual reasoning for complex visual tasks.
- Considerable improvements in comparison with baseline models.
- The experiments with visual grounding benchmarks and multiple VQA benchmarks show the effectiveness.

**Limitations:**

- The results from Table 5 show a limited improvement for the Compile-based Feedback Module. It would be better to add an explanation to this phenomenon.
 - At line 659, it is inappropriate to state that the proposed method outperforms existing models by a large margin. We can see the effectiveness of the proposed method from the evaluation result.

Questions
  - If the VLM returns a feedback that the cropped image does not match the query (e.g., a human face), how to update and improve a specific function in code?
  - For the evaluation of Visual Grounding task (i.e., the RefCOCOs benchmarks), what is the initial codebase used to retrieve the code snippets and then the abstract code extraction?
  - Why do we need a human-in-the-loop process as the feedback module when we expect a model to answer a given question automatically? Could this process help the development of the algorithm?


missing references for MLLMs:
  - Bai, Jinze, et al. "Qwen-vl: A frontier large vision-language model with versatile abilities." arXiv preprint arXiv:2308.12966 (2023).
  - Lu, Haoyu, et al. "DeepSeek-VL: towards real-world vision-language understanding." arXiv preprint arXiv:2403.05525 (2024).
  - Wang, Weihan, et al. "Cogvlm: Visual expert for pretrained language models." arXiv preprint arXiv:2311.03079 (2023).
  - Qi, Ji, et al. "CogCoM: Train Large Vision-Language Models Diving into Details through Chain of Manipulations." arXiv preprint arXiv:2402.04236 (2024).
  - Lin, Ji, et al. "Vila: On pre-training for visual language models." arXiv preprint arXiv:2312.07533 (2023).

**Suitability:**

3

---

### Meta-Review · Area_Chair_Ac92 · 2024-07-05

**Recommendation:** Accept (Oral)
**Confidence:** 5

**Metareview:**

This work proposes a training-free framework that automatically decomposes complex tasks into simpler subtasks and refines programs through auto-feedback. The proposed model is evaluated on image question answering, visual reasoning, and grounding, achieving strong zero-shot performance. After rebuttal, all the reviewers are satisfied with author's response, and give high praise to this work due to the innovative methodology, insightful findings and well-written paper. Therefore, my recommendation is Accept with Oral Presentation, and we encourage the authors to incorporate the reviewers' feedbacks into the revised version.